# Monte Carlo Optimization for Sliding Window Size in Dixon Quality Control of Environmental Monitoring Time Series Data

**Zhongya Fan** [1] **, Huiyun Feng** [2] **, Jingang Jiang** [2,]***, Changjin Zhao** [1] **, Ni Jiang** [1,]***,
**Wencai Wang** [1] **and Fantang Zeng** [1]

[1] State Environmental Protection Key Laboratory of Water Environmental Simulation and Pollution Control, South China Institute of Environmental Sciences, Ministry of Ecology and Environment of PRC, Guangzhou 510530, China; fanzhongya@scies.org (Z.F.); zhaochangjin@scies.org (C.Z.); wangwencai@scies.org (W.W.); zengfantang@scies.org (F.Z.)

[2] Institute of Technical Biology & Agriculture Engineering, Hefei Institutes of Physical Science, Chinese Academy of Sciences, Hefei 230031, China; huiyunf@ipp.ac.cn

* Correspondence: gangzg@ipp.ac.cn (J.J.); jiangni@scies.org (N.J.)

**Abstract:** Outliers are often present in large datasets of water quality monitoring time series data. A method of combining the sliding window technique with Dixon detection criterion for the automatic detection of outliers in time series data is limited by the empirical determination of sliding window sizes. The scientific determination of the optimal sliding window size is very meaningful research work. This paper presents a new Monte Carlo Search Method (MCSM) based on random sampling to optimize the size of the sliding window, which fully takes advantage of computers and statistics. The MCSM was applied in a case study to automatic monitoring data of water quality factors in order to test its validity and usefulness. The results of comparing the accuracy and efficiency of the MCSM show that the new method in this paper is scientific and effective. The experimental results show that, at different sample sizes, the average accuracy is between 58.70% and 75.75%, and the average computation time increase is between 17.09% and 45.53%. In the era of big data in environmental monitoring, the proposed new methods can meet the required accuracy of outlier detection and improve the efficiency of calculation.

**Keywords:** time series environmental monitoring data; Monte Carlo optimization; data quality control; sliding window size

## 1. Introduction

The rapid development of the Internet of Things has promoted the application of smart sensors in the field of the environment, contributing to big data and the multi-dimension characteristics of environmental monitoring [1,2]. Outlier processing is critical in environmental data analysis owing to its significant effect on future analysis and modeling [3,4]. Data quality control requires data mining algorithms, expert experience and Internet thinking [5,6]. Meanwhile, the environment automatically requires monitoring values such as typical time series data, which have a large-scale collection time and include complex causes of outliers. At present, there are many ways to detect outliers in time series, such as outlier detection based on prior rules [7], statistical distribution characteristics [8], the Kalman Filter Model (KLM) and Bayesian model [9], the Generalised Linear Model (GLM) -based algorithm [10], intelligence algorithms [3], etc.

Detection of time series outliers was first studied by Fox (1972) [11,12], who introduced two types of outliers, namely, additive outliers and innovations outliers. It is a common method to determine

time series outliers through the residual analysis of ARMA model [13]. ARMA model requires that time series of analysis must conform to the hypothesis of stationarity, which is difficult to meet in practical application, such as meteorological, hydrological and environmental observation series data. The processing of non-stationary time series to stationary series can be realized by various mathematical transformation methods, such as difference transformation methods, mapping-based transformation methods, and splitting-based transformation methods [14], etc. With the development of datasets outlier detection methods, the detection methods can be divided into the following types: (a) distance-based outlier detection methods [15,16]; (b) density-based outlier detection methods [17]; (c) clustering-based outlier detection methods [18]; (d) sliding window-based outlier detection methods [19]. Environmental monitoring is an important task in today's industrial production, environmental protection, ecology and global climate change research. Advanced and integrated environmental monitoring technology is quite necessary to solve complex ecological and environmental problems, such as remote sensing monitoring technology [20–22], Internet of Things technology [23] and soundscape ecology technology based on sound monitoring [24,25].

The Chinese government has constantly attached great importance to the monitoring and control of environmental pollution [26]. With the deepening of pollution prevention and the great advances in monitoring technology in recent years, continuous automatic in situ monitoring has sent water environment monitoring into a new age. The capability of continuous monitoring in time and space makes environment monitoring technology an essential complementary method for field sampling, laboratory monitoring and analysis. The surface water quality automatic monitoring system is mainly composed of a surface water quality automatic monitoring station (referred to as a water station) and a water quality automatic monitoring data platform (referred to as a data platform). To date, more than 2000 national surface water quality automatic stations have been built in China.

The constructions of large-scale water stations along with the advent of the "Internet +" big data era can meet the demand for higher environmental monitoring and management. With the increasing frequency of water quality monitoring, the amount of data that needs to be processed is growing. Integrating human experience and computer algorithms to effectively extract, select and process the acquired data, recognize the reliability of data, ensure the integrity of information, and reduce the uncertainty of data have become the key challenges which need solutions. A new detection method of time series outliers based on differential analysis and Dixon detection criteria was proposed by Jiang et al. [27] in their previous study and has achieved good application in the time series of marine water element observation. As one of the key parameters in this method, it is urgently required to study and discuss the impact of the size of the sliding detection window on the result of environmental data quality control in great detail, while only empirical suggestions on the range of the parameter are given [27]. The objective of this study is to apply this method to detect the effectiveness of the time series outliers referring to the automatic monitoring of inland water environment. Furthermore, a new algorithm design based on random sampling is developed, realizing the optimal calculation of the sliding window size, and carrying out the experimental verification and comparative analysis. The accuracy and efficiency of the algorithm are quantitatively evaluated according to the experimental data, which is the follow-up to the large-scale data quality of this method. The application of control provides new ideas and effective technical methods for the selection of sliding window parameters of a large number of environmental monitoring data quality control.

## 2. Dixon Quality Control Method

Natural water quality monitoring data in time series has the characteristics of being multi-dimensional and covering long time scales. The quality control and outlier detection of time series are critical for the quality control of environmental monitoring time series data considering the effect of hydrometeors, the electromagnetic environment, and water organisms on the process of data acquisition in addition to the complex natural environment and high monitoring cost. The control means of data quality are mainly realized through regular sensor calibration and maintenance, while

the conventional parallel observation experiments in data quality control are usually difficult to achieve. In this context, Jiang et al. [27] proposed a new detection method by integrating the difference analysis and Dixon detection criteria to account for the outliers in the time observation series of marine water elements. The core assumptions of this method for the quality control of environmental monitoring time series data are as follows:

(1)  In a short period of time, environmental monitoring factors have similar physical and chemical properties;
(2)  the monitoring value in a short time window has the same significance as the parallel repeated observation experiment in outlier detection;
(3)  the time series of environmental monitoring that needs quality control treatment should meet the assumption of stationarity.

How can we meet the requirements of sequence stability? In this method, the first-order or multi-order difference method is used for processing, and the Augmented Dickey-Fuller (ADF) stability test [28,29] is selected to test the stability of the difference time series. For a sequence that does not meet the stability test, the high-order difference transformation is needed to realize the stability of the sequence. Figure 1 is the time sequence diagram and the first-order difference sequence diagram of the original observation values of the experimental data selected in this paper (including the water chlorophyll-a concentration and electronic conductivity). The ADF detection results of the first-order difference sequence meet the test hypothesis of stability.

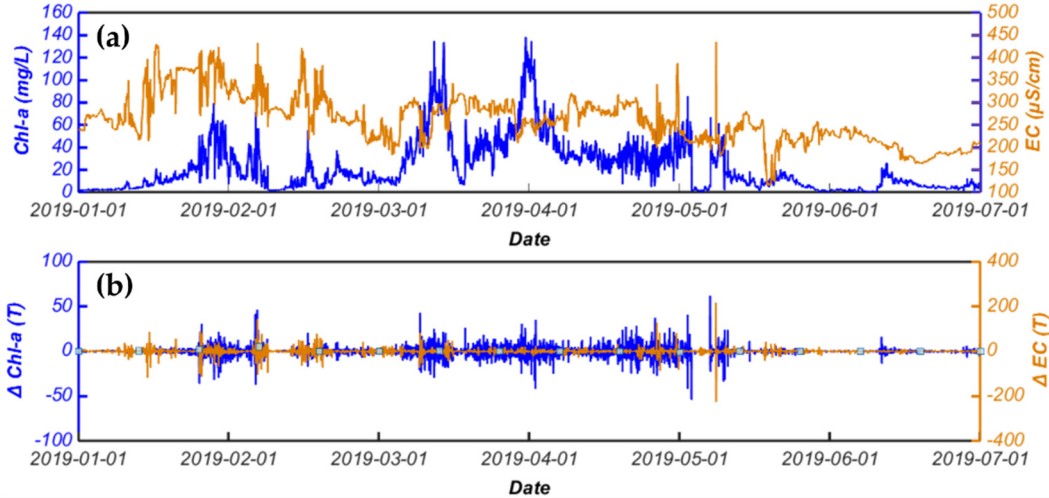

**Figure 1.** Original time series curve and first-order difference series curve of automatic environmental monitoring data. (**a**): The original time series data of the observation series; (**b**): the first-order differential time series data of the observation series. The blue line is the time series data of chlorophyll a concentration observation; the brown line is the time series data of water conductivity observation.

Dixon detection is one of the most commonly used methods for data quality control. It was proposed by the mathematician Dixon in 1950 [30–32] and is a method performed without estimating the mean value and standard deviation. It is used to identify the gross error according to the order difference of measurement data in size order. It is commonly used in consistency testing and in the removal of outlier testing of parallel test measurement data. Here, the parallel observation test is used instead of the statistical window size of the continuous difference sequence; the test calculation method is as follows:

First, sort a set of $w$ differential sequences $\Delta y\,(t+1)$, $\Delta y\,(t+2)$, ... , $\Delta y\,(t+w)$ into $\Delta y\,(t+1)'$, $\Delta y\,(t+2)'$, ... , $\Delta y\,(t+w)'$ from lowest to highest; $t$ is a certain time of time series, $w$ is a statistical inspection window.

Second, consider the lowest and highest value in the ranked data sequence as the questionable values. Identify the lowest or highest value as outliers according to the formula in Table 1.

**Table 1.** Calculation formula of the *Q* value of Dixon detection.

| Sliding Window Size ($w$) | $\Delta y(t+w)'$ Is a Dubious Value | $\Delta y(t+1)'$ Is a Dubious Value |
|:---:|:---:|:---:|
| 3–7 | $Q_{10} = \dfrac{\Delta y(t+w)' - \Delta y(t+w-1)'}{\Delta y(t+w)' - \Delta y(t+1)'}$ | $Q_{10} = \dfrac{\Delta y(t+2)' - \Delta y(t+1)'}{\Delta y(t+w)' - \Delta y(t+1)'}$ |
| 8–10 | $Q_{11} = \dfrac{\Delta y(t+w)' - \Delta y(t+w-1)'}{\Delta y(t+w)' - \Delta y(t+2)'}$ | $Q_{11} = \dfrac{\Delta y(t+2)' - \Delta y(t+1)'}{\Delta y(t+w-1)' - \Delta y(t+1)'}$ |
| 11–13 | $Q_{21} = \dfrac{\Delta y(t+w)' - \Delta y(t+w-2)'}{\Delta y(t+w)' - \Delta y(t+2)'}$ | $Q_{21} = \dfrac{\Delta y(t+3)' - \Delta y(t+1)'}{\Delta y(t+w-1)' - \Delta y(t+1)'}$ |
| 14–30 | $Q_{22} = \dfrac{\Delta y(t+w)' - \Delta y(t+w-2)'}{\Delta y(t+w)' - \Delta y(t+3)'}$ | $Q_{22} = \dfrac{\Delta y(t+3)' - \Delta y(t+1)'}{\Delta y(t+w-2)' - \Delta y(t+1)'}$ |

After calculating the value of *Q*, the threshold $Q_\alpha$ is read from Table 2 according to the selected significance level $\alpha$ and sliding window size *w*. The decision to retain or eliminate the dubious data is made according to the following criteria:

(1)　If $Q > Q_{0.01}$, then the dubious values are outliers that must processed;
(2)　If $Q_{0.05} < Q < Q_{0.01}$, then the dubious values are deviant values that can be retained or processed;
(3)　If $Q < Q_{0.05}$, then the dubious values are normal values and should be retained;
(4)　If it is necessary to address a dubious value, the median of the remaining valid data is commonly used in lieu of the abnormal data. However, an appropriate interpolation method such as spline interpolation may be more reasonable. In this paper, the inverse distance weighted average method is used instead of the median.

**Table 2.** Decision table for Dixon criterion detection threshold.

| Significance Level | $w$ | | | | | | | | | | | | | | |
|:---:|:---:|:---:|:---:|:---:|:---:|:---:|:---:|:---:|:---:|:---:|:---:|:---:|:---:|:---:|:---:|
| | 3 | 4 | 5 | 6 | 7 | 8 | 9 | 10 | 11 | 12 | 13 | 14 | 15 | 16 |
| $Q_{0.05}$ | 0.941 | 0.765 | 0.642 | 0.560 | 0.507 | 0.554 | 0.512 | 0.477 | 0.576 | 0.546 | 0.521 | 0.546 | 0.525 | 0.507 |
| $Q_{0.01}$ | 0.988 | 0.889 | 0.780 | 0.698 | 0.637 | 0.683 | 0.635 | 0.597 | 0.679 | 0.642 | 0.615 | 0.641 | 0.616 | 0.595 |

| Significance Level | $w$ | | | | | | | | | | | | | | |
|:---:|:---:|:---:|:---:|:---:|:---:|:---:|:---:|:---:|:---:|:---:|:---:|:---:|:---:|:---:|:---:|
| | 17 | 18 | 19 | 20 | 21 | 22 | 23 | 24 | 25 | 26 | 27 | 28 | 29 | 30 |
| $Q_{0.05}$ | 0.490 | 0.475 | 0.462 | 0.450 | 0.440 | 0.431 | 0.422 | 0.413 | 0.406 | 0.399 | 0.393 | 0.387 | 0.381 | 0.376 |
| $Q_{0.01}$ | 0.577 | 0.561 | 0.547 | 0.535 | 0.526 | 0.516 | 0.507 | 0.497 | 0.489 | 0.482 | 0.474 | 0.468 | 0.462 | 0.456 |

The detection of all data in the time series is carried out in the form of window sliding. As one of the key parameters of the method, the size of the detection window has not been analyzed and researched in-depth in the literature [27]; only empirical suggestions on the range of the value are given.

## 3. Optimization Algorithms for Sliding Window Size

Dixon detection can realize simple and efficient outlier detection by calculating the *Q* value and decision table when detecting small amounts of sample data [33]. The method used by Jiang et al. [27] realizes the detection application of a large sample size for long-term time series data through window sliding. In order to set the size of the sliding window reasonably and make the efficiency of outlier detection optimal, a new Monte Carlo algorithm is proposed in this paper. The new algorithm makes use of the advantages of computer random sampling and realizes the determination of the optimal sliding window size through the evaluation standard of detection rate. In order to quantitatively evaluate the accuracy and efficiency of the new algorithm, we selected the measured water quality time series data from the automatic monitoring station for the experimental verification and comparative analysis of the new algorithm.

Considering that the probability of abnormal data is usually very small, the number of samples for random sampling to determine the optimal sliding window is required to be large enough so that the calculation results can meet the accuracy of the algorithm. Obviously, the larger the sample size, the lower the efficiency of the algorithm. In the design of the algorithm, it is recommended to select $[0.5n, 0.8n]$ ($n$ is the length of time series record) as the length for random sampling calculation during random sample selection. The algorithm flow and schematic diagram are shown in Figure 2.

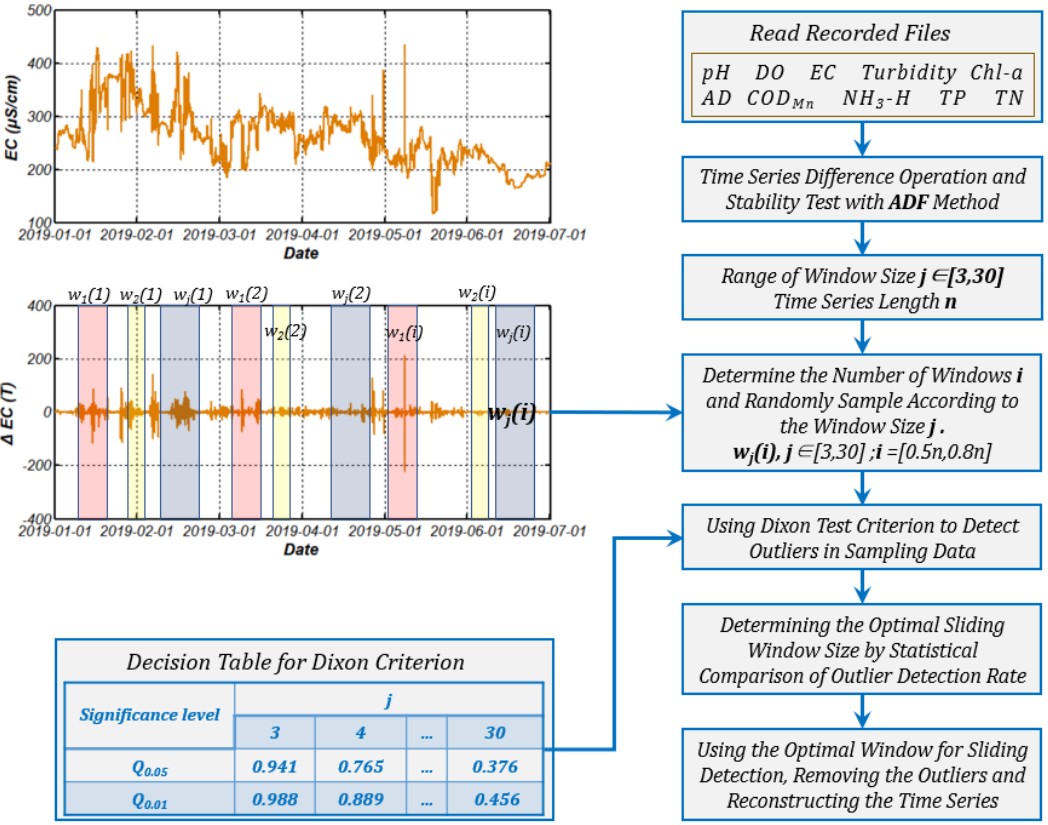

**Figure 2.** Flow chart and schematic diagram of the Monte Carlo optimization algorithm. Top left subgraph: water conductivity as a case to illustrate the sampling process of random window; bottom left subgraph: Dixon detection decision table; right subgraph: flow chart to determine the optimal window based on the random sampling algorithm.

The determination criterion of the optimal window is determined by the maximum detection ratio of outlier data. The specific calculation is as follows:

$$\begin{cases} odr(J,i) = max\left(\frac{Y_j(i)}{n}\right), \ j \in [3,30]; i = [0.5n, 0.8n] \\ ow = J \end{cases} \tag{1}$$

where $n$ is the number of samples recorded in the time series, $j$ is the number of windows, $i$ is the number of randomly generated samples, $Y_j(i)$ is the number of abnormal points under the random sampling window $w_j(i)$, $odr(J,i)$ is the maximum abnormal detection rate of random sampling, and $ow$ is the optimal detection window, corresponding to the window number $J$ of the maximum abnormal detection rate.

To facilitate the understanding of the algorithm, the pseudo code structure of the algorithm is as follows. The program of MCSM algorithm will be available in the open code. One can find the link in the Supplementary Materials.

---

**Algorithm 1: Monte Carlo Search Method (MCSM).**

---

**Input:**

*P*-Path to read file
$Q_\alpha$-Decision table for Dixon criterion detection
*j*-Range of window size, $j \in [3,30]$
*i*-Randomly sampling scale
*k*-Number of random simulations

**Preprocess:**

*n*-Calculate differential time series length
*dx*-Differential operation on time series
*H*-Using *adftset* function to test the stationarity of difference series, H = 1: Stationary series; H = 0: Nonstationary series

**Main Procedure:**
　　*for* *kk=1:k*
　　　　*for* *ii=1:i*
　　　　　　*for* *jj=3:30*
　　　　　　　　*do* Calculate the *Q* value according to the formula in Table 1 and using Dixon criterion to detect outliers
　　　　　　　　*end*
　　　　　　*end*
　　　　*end* *odr(J,i)* Calculate maximum detection ratio of outlier data according to Equation (1)
　　**Output:**
　　*ow = J*

---

## 4. Experimental Studies

### 4.1. Data Sources

In order to test the validity and accuracy characteristics of the new algorithm, the representative monitoring data of the water quality automatic monitoring station, which is located in Huayang Lake Group in the middle reaches of the Yangtze River, were selected as the test data. The data acquisition time was the latest recorded data from January 2019 to July 2019. The water quality factor parameter information of the algorithm test data is shown in Table 3. It can be seen from the information in Table 3 that the selected water quality parameter factors included conventional physical and chemical factors and biological parameter factors. Among these parameter factors, the recording period of pH, dissolved oxygen (DO), electronic conductivity (EC), turbidity, chlorophyll-a (Chl-a) and algal density (AD) water quality factors was every 1 hour, and the length of the sequence was 4723, while the recording period of the permanganate index ($COD_{MN}$), ammonia–nitrogen ($NH_3$-N), total phosphorus (TP), and total nitrogen (TN) water quality factors is was 4 hours, and the length of the sequence was 1187. Among these environmental monitoring factors, the factors with large coefficient of variation include NH3-N, AD, turbidity and Chl-a.

### 4.2. Experiment Platform and Initial Conditions

Matlab 2016b 64-bit software was selected as the program development platform of the new algorithm, and the computer running the program was a Huawei Honor Magic Book. The CPU of the computer was AMD Ryzen 5 2500U (at 2.00 GHz), the RAM was 8.00 GB, and the operating system was Window 10.

**Table 3.** Information about the water quality factor data sources.

| Water Factors | Sampling Period | Time Series Length | Instrument Accuracy | Unit | Mean Value | Standard Deviation | Coefficient of Variation |
|---|---|---|---|---|---|---|---|
| pH | 1 h | 4723 | ±0.2 pH | – | 8.29 | 0.73 | 0.09 |
| Dissolved Oxygen | 1 h | 4723 | ±0.2 mg/L (≤20 mg/L) ±0.6 mg/L (>20 mg/L) | mg/L | 10.43 | 3.02 | 0.29 |
| Electronic Conductivity | 1 h | 4723 | ±0.01% | μS/cm | 261.00 | 53.90 | 0.21 |
| Turbidity | 1 h | 4723 | ±0.01NTU | NTU (Nephelometric Turbidity Unit) | 99.20 | 118.51 | 1.19 |
| Chlorophyll-a | 1 h | 4723 | ±3% | μg/L | 21.81 | 22.46 | 1.03 |
| Algal Density | 1 h | 4723 | ±3% | cells/ml | 14,906 | 19,221 | 1.29 |
| Permanganate Index | 4 h | 1187 | ±5% | mg/L | 8.80 | 4.86 | 0.55 |
| Ammonia-Nitrogen | 4 h | 1187 | ±10% | mg/L | 0.12 | 0.17 | 1.42 |
| Total Phosphorus | 4 h | 1187 | ±10% | mg/L | 0.17 | 0.11 | 0.65 |
| Total Nitrogen | 4 h | 1187 | ±10% | mg/L | 1.37 | 0.65 | 0.47 |

The initial conditions of the algorithm were as follows: (a) the window scale $w$ was defined as 5–30; (b) the random sampling scale was set as $0.5n$, $0.6n$, $0.7n$ and $0.8n$, respectively, where $n$ is the length of the analysis sequence (4723 and 1187 in this paper); (c) the characteristic of the random algorithm is randomness, and the result of a single calculation is often randomness. In order to make the experimental results statistically significant, the number of repeated simulation experiments was defined as 40.

*4.3. Results and Comparative Analysis*

The Full Time Series Sliding Search Method (FTSSSM) was used to obtain the correct window size. In order to scientifically compare the correctness of the new Monte Carlo Search Method (MCSM), FTSSSM experiments were carried out at the same time. The evaluation indexes of the calculation results of FTSSSM and MCSM included the running time of the program and the accuracy of the optimal window.

We carried out the experiment according to the experimental platform and initial conditions in Section 4.2. The statistical information of the experimental calculation results is shown in Table 4. It can be seen from Table 4 that the optimal window size obtained from the FTSSSM experiments was between 8 and 18, and there were two or three optimal windows for DO and turbidity. With the increase of sampling scale, the accuracy of MCSM continued to improve, and the running time of the program continued to increase. When the sampling scale was $0.8n$, the optimal window accuracy of different water quality factors was between 67.5% and 85%. When the sampling scale was $0.5n$, the optimal window accuracy of different water quality factors was between 52.5% and 65%.

In order to quantitatively analyze the improvement of the running time efficiency of MCSM, the running time of different sampling scales was selected and compared with that of FTSSSM experiments, and we calculated the rate of computation time increase. The rate of computation time increase was as follows:

$$rr = \frac{t_j(n) - t_j(i)}{t_j(n)} \times 100\%, j \; from \; 5 \; to \; 30 \tag{2}$$

where *rr* is the rate of computation time increase, *j* is the number of windows (traverse calculation from 3 to 30), $t_j(n)$ is the run-time of FTSSSM, and $t_j(i)$ is the run-time of MCSM.

**Table 4.** Statistical information table of experimental calculation results.

| Water Factors | Full Time Series Sliding Search Method | | Monte Carlo Search Method | | | | | | | |
| | | | 0.5n | | 0.6n | | 0.7n | | 0.8n | |
| | Run Time (s) | Optimal Sliding Window Size | Run Time (s) | Optimal Window Accuracy (%) | Run Time (s) | Optimal Window Accuracy (%) | Run Time (s) | Optimal Window Accuracy (%) | Run Time (s) | Optimal Window Accuracy (%) |
|---|---|---|---|---|---|---|---|---|---|---|
| pH | 1.0508 | 14 | 0.5950 | 62 | 0.6023 | 65 | 0.6949 | 70 | 0.8867 | 77.5 |
| Dissolved Oxygen | 1.0701 | 12, 14 | 0.5687 | 65 | 0.6527 | 72.5 | 0.8605 | 77.5 | 0.9332 | 80 |
| Electronic Conductivity | 1.1270 | 16 | 0.6074 | 62.5 | 0.7332 | 67.5 | 0.7504 | 75 | 1.0313 | 85 |
| Turbidity | 1.1163 | 16, 17, 18 | 0.6113 | 60 | 0.6648 | 67.5 | 0.8070 | 72.5 | 0.9176 | 80 |
| Chlorophyll-a | 1.1410 | 17 | 0.5352 | 57.5 | 0.7680 | 65 | 0.8770 | 67.5 | 0.9316 | 75 |
| Algal Density | 1.1251 | 14 | 0.6469 | 55 | 0.6754 | 65 | 0.7352 | 72.5 | 0.9437 | 75 |
| Permanganate Index | 0.2606 | 9 | 0.1445 | 57.5 | 0.1738 | 62.5 | 0.1879 | 65 | 0.1984 | 75 |
| Ammonia–Nitrogen | 0.2658 | 8 | 0.1574 | 52.5 | 0.1789 | 60 | 0.1816 | 67.5 | 0.2242 | 70 |
| Total Phosphorus | 0.2426 | 12 | 0.1418 | 55 | 0.1637 | 57.5 | 0.1703 | 65 | 0.2039 | 67.5 |
| Total Nitrogen | 0.2586 | 14 | 0.1262 | 60 | 0.1477 | 62.5 | 0.1723 | 65 | 0.1906 | 72.5 |

Figure 3 shows the rate of computation time increase histogram calculated by Equation (2). It can be seen from Figure 3 and Table 4 that the time series of water quality factors with the highest rate of computation time increase was the Chl-a observation series. When the random sampling scale was $i = 0.5n$, the maximum rate of computation time increase was 53.09%, and the optimal window accuracy was 57.5%. The time series of the water quality factor with the lowest rate of computation time increase was the EC observation series. When the random sampling scale was $i = 0.8n$, the rate of the computation time increase was 8.49%, and the optimal window accuracy was 80%. The statistical results show that, when the random sampling scale was $i = 0.5n$, the average rate of computation time increase with all water quality factors was 45.53%; when the random sampling scale was $i = 0.6n$, the average rate was 37.11%; when the random sampling scale was $i = 0.7n$, the average rate was 29.51%; and when the random sampling scale was $i = 0.8n$, the average rate was 17.09%.

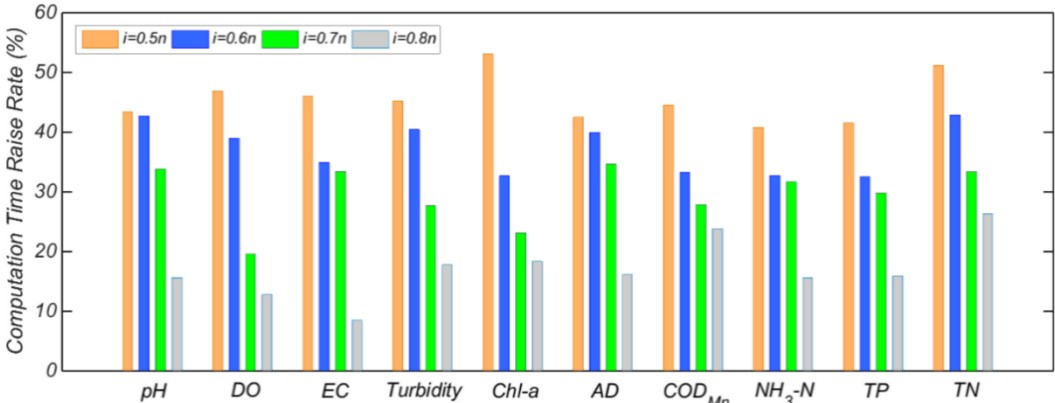

**Figure 3.** Histogram of the rate of computation time increase under different sampling scales.

Taking the turbidity time series data as an example, the results in Figure 4 show the statistical characteristics of MCSM and FTSSSM under different sampling scales. The curve results of FTSSSM show that the outlier detection rate showed a rising trend first and then declined (red line in Figure 4a). The optimal window sizes of the turbidity time series data were 16, 17 and 18, under which 5.99% of the outlier data could be detected. The average outlier detection rate curve of the new algorithm proposed in this paper was basically consistent with the red line under different sampling scales. Because the scale of random sampling did not include all the data of the sequences, the outlier detection rate was lower than the red line. It should be noted that the goal of the new algorithm was to determine the optimal window size, and after the optimal window size was determined, the sliding detection of the whole sequence was still achieved through the optimal window size.

The detailed data in Figure 4b shows that when $i = 0.5n$, a total of seven optimal window sizes were obtained, which were 14, 15, 16, 17, 18, 19 and 20. Among the seven results, the window sizes 14, 15, 19 and 20 were wrong. The largest proportion of window sizes was 19, and its proportion value was 25%; the smallest proportion of window sizes was 14, and its proportion value 2.5%. When $i = 0.6n$, six optimal window sizes were calculated, which were 15, 16, 17, 18, 19 and 20. The largest proportion of window sizes was 17, and its proportion value was 30%; the smallest proportion of window sizes was 20, and its proportion value was 2.5%. When $i = 0.7n$, six optimal window sizes were calculated, which were 15, 16, 17, 18, 19 and 20. The largest proportion of window sizes was 18, and its proportion value was 32.5%; the smallest proportion of window sizes was 20, and its proportion value was 2.5%. When $i = 0.8n$, four optimal window sizes were calculated, which were 16, 17, 18 and 19. The largest proportion of window sizes was 18, and its proportion value was 37.5%; the smallest proportion of window sizes was 16, and its proportion value was 17.5%.

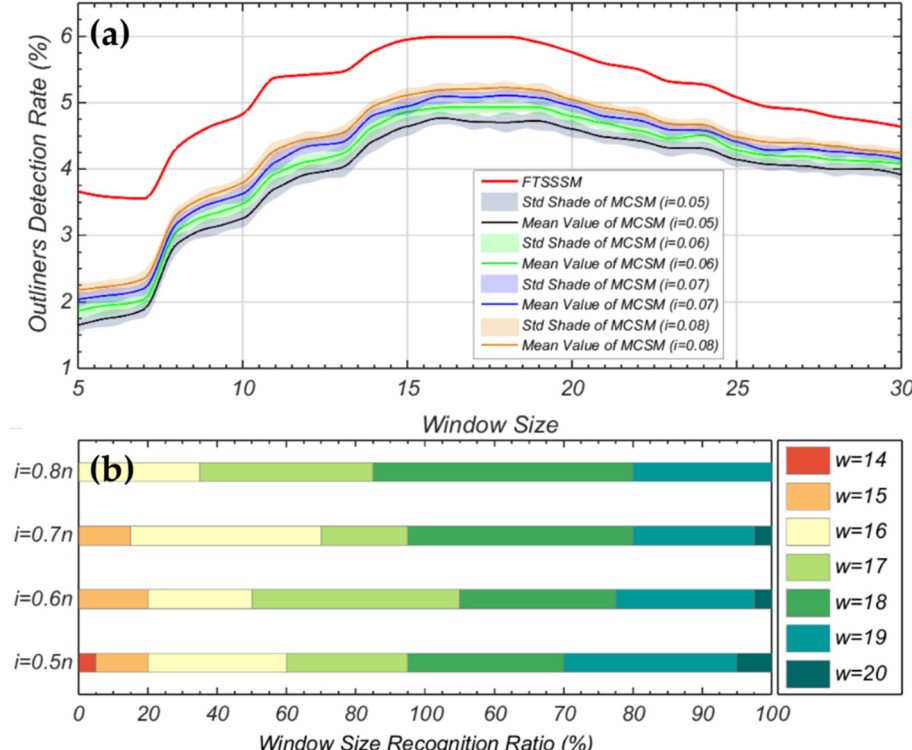

**Figure 4.** Statistical characteristics of Monte Carlo Search Method (MCSM) and Full Time Series Sliding Search Method (FTSSSM) results under different sampling scales. (**a**) The red line indicates the outlier detection rate of FTSSSM under different window sizes; the blue–black line indicates the outlier detection rate of MCSM under different window sizes with a sampling scale of $i = 0.5n$, and the blue–black shadow is the standard deviation area of 40 test results; the green line indicates the outlier detection rate of MCSM under different window sizes with a sampling scale of $i = 0.6n$, and the green shadow is the standard deviation area of 40 test results; the blue line indicates the outlier detection rate of MCSM under different window sizes with a sampling scale of $i = 0.7n$, and the blue shadow is the standard deviation area of 40 test results; the brown line indicates the outlier detection rate of MCSM under different window sizes with a sampling scale of $i = 0.8n$, and the brown shadow is the standard deviation area of 40 test results. (**b**) The results and percentage distribution of optimal window under different sampling scales.

Figure 4b also shows the following regular pattern, and in the calculation results, the correct results and the incorrect results exist together. The results of the incorrect window sizes were near the optimal window sizes. In view of this regular pattern, we gathered statistics on the outlier detection rate corresponding to the optimal window sizes, the suboptimal window sizes and the worst window sizes, and the results are shown in Table 5. It can be seen from this table that the suboptimal window sizes were generally distributed near the optimal window sizes. The difference between the outlier detection rates of the suboptimal window sizes and the optimal window sizes was very small, while the difference between the worst window sizes and the optimal window sizes was obvious.

Taking EC and Chl-a time series data as examples, the outlier detection results under the optimal window size sliding detection conditions are shown in Figure 5. This shows that the combination of sliding windows and Dixon detection criteria can well detect the outliers of observation data. The reconstructed observation time series of water quality factors only dealt with the outlier data, while the normal values in the series were effectively preserved, and the features of the original observation data were retained to a great extent.

**Table 5.** Comparison information of different representative window sizes and anomaly detection rates.

| Water Factors | Optimal Window Sizes | | Suboptimal Window Sizes | | Worst Window Sizes | |
|---|---|---|---|---|---|---|
| | Window Size | Outliers Detection Rate (%) | Window Size | Outliers Detection Rate (%) | Window Size | Outliers Detection Rate (%) |
| **pH** | 14 | 5.25 | 12 | 5.23 | 23 | 3.01 |
| **Dissolved Oxygen** | 12,14 | 3.09 | 11 | 2.99 | 29 | 1.80 |
| **Electronic Conductivity** | 16 | 6.69 | 15 | 6.58 | 5 | 3.47 |
| **Turbidity** | 16, 17, 18 | 5.99 | 15 | 5.95 | 7 | 3.56 |
| **Chlorophyll-a** | 17 | 3.05 | 16 | 3.02 | 6 | 2.10 |
| **Algal Density** | 14 | 3.22 | 16 | 2.90 | 7 | 2.10 |
| **Permanganate Index** | 9 | 6.23 | 12 | 5.81 | 30 | 4.04 |
| **Ammonia–Nitrogen** | 8 | 12.47 | 14 | 12.13 | 28 | 6.66 |
| **Total Phosphorus** | 12 | 7.16 | 14 | 6.99 | 5 | 4.04 |
| **Total Nitrogen** | 14 | 7.92 | 9 | 7.58 | 30 | 4.80 |

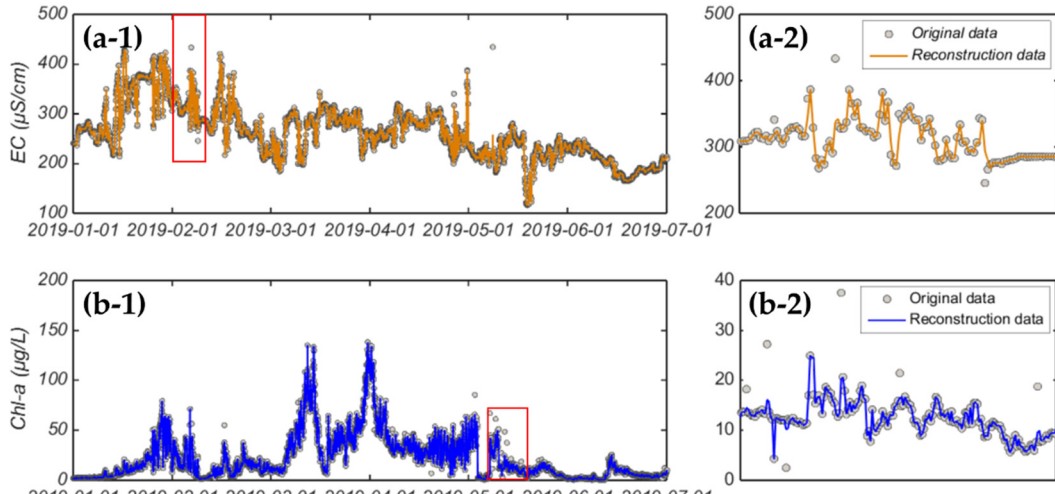

**Figure 5.** Detection of outliers and sequence reconstruction of electronic conductivity (EC) and chlorophyll-a (Chl-a) time series by optimal window size sliding. (**a-1**): The original observation time series data and reconstruction time series data of EC; (**a-2**): Local curve of red box in the figure (**a-1**); (**b-1**): The original observation time series data and reconstruction time series data of Chl-a; (**b-2**): Local curve of the red box in the figure (**b-1**).

## 5. Discussion and Conclusions

Through the experiment and analysis in this paper, we find that the method proposed by Jiang et al. [27] is very effective in the detection of outliers and quality control of time series data in water environment monitoring. Using the key parameter of the detection method, the size of the sliding detection window, through the experiment and analysis in our work, the new method is simple and effective. After a thorough analysis of the relationship between sliding window size and detection accuracy, it can be found that there is a complex multidimensional logical correspondence between them, which makes it difficult to establish a quantitative mathematical model. In dealing with these problems, the Monte Carlo method is undoubtedly the best method and can fully take advantage of computers and statistics. Because of the advantages of the Monte Carlo method, it has been widely used in many different fields [34–36]. The Monte Carlo method also has its shortcomings, the most prominent of which is that it has random characteristics in simulation modeling, the simulation results are statistically significant, and each independent simulation result is uncertain. According to the experimental results in this paper, the accuracy and efficiency of the calculation results are different in different sampling scales and optimal window sizes. Accuracy and calculation efficiency cannot be

satisfied at the same time: to improve accuracy, efficiency must be sacrificed; to improve efficiency, accuracy must be sacrificed.

The experimental results in this paper support the following conclusions: when $i = 0.5n$, the average calculation efficiency is increased by 45.53% and the average accuracy is 58.70%; when $i = 0.6n$, the average calculation efficiency is increased by 37.11% and the average accuracy is 64.5%; when $i = 0.7n$, the average calculation efficiency is increased by 29.51% and the average accuracy is 69.75%; when $i = 0.8n$, the average calculation efficiency is increased by 17.09% and the average accuracy is 75.75%.

The accuracy of the optimal window sizes is statistically significant, which is the characteristic of the algorithm itself. The results in Figure 4 and Table 5 indicate that even if the MCSM does not find the optimal window sizes, its recommended window sizes are also distributed near the optimal window sizes. In this case, we recommend the suboptimal window sizes, because they do not have a big impact on the final sequence outlier detection rate.

Afore-mentioned experimental results also explain the obvious disadvantage of the new method is that the calculation accuracy and efficiency can't be satisfied at the same time. With the rapid development and progress of computer computing capacity and artificial intelligence technology, it is an inevitable trend to apply the new method to the parameter optimization algorithm of sliding window size. For the exploration of new methods, we will continue to carry out in-depth research in the future work.

**Supplementary Materials:** The Matlab program of MCSM algorithm is available online at: http://www.mdpi.com/2076-3417/10/5/1876/s1. Welcome readers who are interested to use it as scientific research purposes. Supplementary material in the folder contains MCSM to determine the size of the sliding window program (DixonQC_MCSM.m) and Dixon quality control method program (DixonQC.m). The folder also contains two water quality time series monitoring data files for program testing (testdata1.xlsx and testdata2.xlsx). The program files provided are required to be run in the Matlab platform.

**Author Contributions:** Conceptualization, J.J. and N.J.; methodology, J.J.; software, J.J. and N.J.; validation, C.Z. and W.W.; formal analysis, Z.F.; investigation, N.J.; resources, W.W.; data curation, N.J. and W.W.; writing—original draft, Z.F. and N.J.; writing—review and editing, H.F. and J.J.; visualization, J.J.; supervision, F.Z.; project administration, Z.F.; funding acquisition, H.F. and F.Z. All authors have read and agreed to the published version of the manuscript.

**Funding:** This research was funded by Key-Area Research and Development Program of Guangdong Province, grant number 2019B110205003, National Science and Technology Major Project for Water Pollution Control and Treatment, grant number 2017ZX07603-005,Science and Technology Service Network Initiative, Chinese Academy of Sciences, grant number KFJ-STS-QYZD-173, and National Yangtze River Conservation and Restoration Joint Research Project, grant number 2018CJA030301-014.

**Acknowledgments:** The authors would like to specially thank the Anqing Ecological Environment Protection Bureau and Susong Ecological Environment Protection Bureau for providing the valuable water quality monitoring data of automatic monitoring station.

**Conflicts of Interest:** The authors declare no conflict of interest.

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
