# Peer review of "Monte Carlo Optimization for Sliding Window Size in Dixon Quality Control of Environmental Monitoring Time Series Data"

_applsci, doi:10.3390/app10051876_

Round 1

Reviewer 1 Report

The authors describe a Monte Carlo algorithm to analyse the correct window size to detect outliers in time series of environmental data. The theoretical and experimental part are well presented. I am not a mathematician, but the description seems correct. The described examples show the advantage of the algorithm compared to the previously reported methods.   

It would be helpful for other groups if the Matlab source code could be made available in the supplemental section. 

Author Response

Dear Professor

   Thank you very much for your review of our manuscript entitled “Monte Carlo Optimization for Sliding Window Size in Dixon Quality Control of Environmental Monitoring Time Series Data” (Manuscript ID: applsci-722945).

   We also sincerely thank you for your recognition of our work. We hope that our work and program can be helpful to the peers. Therefore, we are willing to shared our program code on MDPI publishers platform. The main file information in the program folder is shown in figure 1

Figure1 Main file information in the program folder

1)This folder mainly contains DixonQCow.m and DixonQC.m program files;

2)Program files need to run on MATLAB platform;

3)DixonQCow.m is an algorithm program for optimal window size optimization;

4)DixonQC.m is a program for outlier detection by Dixio method;

5)testdata1.xlsx and testdata2.xlsx test data of the program;

6)  For detailed algorithm principle, please refer to references:"Monte Carlo Optimization for Sliding Window Size in Dixon Quality Control of Environmental Monitoring Time Series Data"      and " Outlier detection and sequence reconstruction in continuous time series of ocean observation data based on difference analysis and the Dixon criterion";

7) The contact information of the program developer is gangzg@ipp.ac.cn .

Enjoy!

Thank you very much for your comments.

Sincerely yours,

Ni Jiang

Institution:  National Key Laboratory of Water Environmental Simulation and Pollution Control, South China Institute of Environmental Sciences, Ministry of Ecology and Environment

E-mail:jiangni@scies.org

Reviewer 2 Report

This article presents a study to optimize the size of sliding windows to allow automatic detection of outliers in environmental parameters. For this purpose, a new MCSM method is proposed and applied to the study of certain water quality factors.

As the authors themselves refer to, this work is the continuation and culmination of a previous study on "outlier detection and sequence reconstruction in continuous time series of ocean observation data based on difference analysis and the Dixon criterion"; where it remains to optimize window sizes to obtain more stable and solid results.

The paper is well structured, introduces well the concepts, the previous studies on which it is based and the objectives it aims to achieve. The methodology, the experiments performed on real data and the results are rigorous and well described. From a formal point of view, I consider it complete and well executed.

However, from the point of view of scientific value, I would like to make some comments in case the authors can improve the current work or future lines of work.

1) I think it would be interesting to highlight the main contribution made and to put it into value. What is new about it compared to other studies?

2) The results obtained are moderate (average accuracy between 58.7% and 75.7%). Has it been compared with other studies or methods? This comparison could be interesting. It could be interesting to use Machine Learning techniques that give very good results. I suggest that you consult, for example, the following paper, where results above 80% are achieved. (https://www.mdpi.com/1424-8220/18/6/1803 ).

3) It would be interesting to describe the sample sizes that have been worked with. The total records processed to put in value the tests performed. In general I think that a good, structured and well-developed work has been done; however, I do not finish seeing the scientific novelty, I consider it not very relevant.

Some minor aspects: I suggest leaving a blank line of space before and after the tables and some of the figures, they are very close to the text. For example:

Line 115: leave a blank line between 114 and 115. Also leave another free blank before 116.

Line 228: leave a blank line before it.

Line 257: leave a blank line before the figure. Same as Table 5 and Figure 5.

Author Response

Dear Professor

Thank you for your letter and for the reviewer’s comments on our manuscript entitled “Monte Carlo Optimization for Sliding Window Size in Dixon Quality Control of Environmental Monitoring Time Series Data” (Manuscript ID: applsci-722945). These comments have important guiding significance for our research and reference value for the revision and improvement of our article. We have studied comments carefully and made corrections hoping for approval. Revised portion are marked in red in the paper. The main revision in the paper and the responds to the reviewer’s comments are as flowing:

Responds to the reviewer’s comments:

COMMENT:1) I think it would be interesting to highlight the main contribution made and to put it into value. What is new about it compared to other studies?

RESPONSE: At present, the technology of environmental monitoring has been continuously developed, and the amount of environmental monitoring data is increasing. Data quality control is an important part of data processing and analysis. In view of the importance of environmental monitoring data quality control, many peers have done a lot of meaningful work, and our work is also based on the development and continuation of previous work. In order to illustrate the significance and novelty of our work, we make a further summary of the previous work in the introduction part of the article.

The introduction adds the following:

“Detection of time series outliers was first studied by Fox (1972) [11,12], who introduced two types of outliers, namely, additive outliers and innovations outliers. It is a common method to determine time series outliers through the residual analysis of ARMA model [13]. ARMA model requires that time series of analysis must conform to the hypothesis of stationarity, which is difficult to meet in practical application, such as meteorological, hydrological and environmental observation series data. The processing of non-stationary time series to stationary series can be realized by various mathematical transformation methods, such as difference transformation methods, mapping-based transformation methods, and splitting-based transformation methods [14], etc. With the development of datasets outlier detection methods, the detection methods can be divided into the following types: a) Distance-based outlier detection methods [15,16]; b) Density-based outlier detection methods [17]; c) Clustering-based outlier detection methods [18]; d) Sliding window-based outlier detection methods [19]. Environmental monitoring is an important task in today's industrial production, environmental protection, ecology and global climate change research. Advanced and integrated environmental monitoring technology is quite necessary to solve complex ecological and environmental problems, such as remote sensing monitoring technology [20-22], internet of things technology [23] and soundscape ecology technology based on sound monitoring [24,25].”

The newly added references are as follows:

  1. Fox, A.J. Outliers in Time Series. Journal of the Royal Statistical Society 1972, 34, 350-363, doi:10.1111/j.2517-6161.1972.tb00912.
  2. Choy, K. Outlier detection for stationary time series. Journal of Statistical Planning and Inference 2001, 99, 111-127, doi:10.1016/S0378-3758(01)00081-7.
  3. Arumugam, P.; Saranya, R. Outlier Detection and Missing Value in Seasonal ARIMA Model Using Rainfall Data*. Materials Today: Proceedings 2018, 5, 1791-1799, doi:10.1016/j.matpr.2017.11.277.
  4. Salles, R.; Belloze, K.; Porto, F.; Gonzalez, P.H.; Ogasawara, E. Nonstationary time series transformation methods: An experimental review. Knowledge-Based Systems 2019, 164, 274-291, doi:10.1016/j.knosys.2018.10.041.
  5. Giménez, E.; Crespi, M.; Garrido, M.S.; Gil, A.J. Multivariate outlier detection based on robust computation of Mahalanobis distances. Application to positioning assisted by RTK GNSS Networks. International Journal of Applied Earth Observation and Geoinformation 2012, 16, 94-100, doi:10.1016/j.jag.2011.11.011.
  6. Angiulli, F.; Basta, S.; Lodi, S.; Sartori, C. Reducing distance computations for distance-based outliers. Expert Systems with Applications 2020, 147, 113215, doi:10.1016/j.eswa.2020.113215.
  7. Tang, B.; He, H. A local density-based approach for outlier detection. Neurocomputing 2017, 241, 171-180, doi:10.1016/j.neucom.2017.02.039.
  8. Christy, A.; Gandhi, G.M.; Vaithyasubramanian, S. Cluster Based Outlier Detection Algorithm for Healthcare Data. Procedia Computer Science 2015, 50, 209-215, doi:10.1016/j.procs.2015.04.058.
  9. Wang, B.; Yang, X.-C.; Wang, G.-R.; Yu, G. Outlier Detection over Sliding Windows for Probabilistic Data Streams. Journal of Computer Science and Technology 2010, 25, 389-400, doi:10.1007/s11390-010-9332-2.
  10. Bauer, M.E. Remote Sensing of Environment: History, Philosophy, Approach and Contributions, 1969 –2019. Remote Sensing of Environment 2020, 237, 111522, doi:10.1016/j.rse.2019.111522.
  11. de Araujo Barbosa, C.C.; Atkinson, P.M.; Dearing, J.A. Remote sensing of ecosystem services: A systematic review. Ecological Indicators 2015, 52, 430-443, doi:10.1016/j.ecolind.2015.01.007.
  12. Turner, W.; Spector, S.; Gardiner, N.; Fladeland, M.; Sterling, E.; Steininger, M. Remote sensing for biodiversity science and conservation. Trends in Ecology & Evolution 2003, 18, 306-314, doi:10.1016/S0169-5347(03)00070-3.
  13. Malche, T.; Maheshwary, P.; Kumar, R. Environmental Monitoring System for Smart City Based on Secure Internet of Things (IoT) Architecture. Wireless Personal Communications 2019, 107, 2143-2172, doi:10.1007/s11277-019-06376-0.
  14. Pijanowski, B.C.; Farina, A.; Gage, S.H.; Dumyahn, S.L.; Krause, B.L. What is soundscape ecology? An introduction and overview of an emerging new science. Landscape Ecology 2011, 26, 1213-1232, doi:10.1007/s10980-011-9600-8.
  15. Luque, A.; Gómez-Bellido, J.; Carrasco, A.; Barbancho, J. Optimal Representation of Anuran Call Spectrum in Environmental Monitoring Systems Using Wireless Sensor Networks. Sensors 2018, 18, doi:10.3390/s18061803.

COMMENT:2) The results obtained are moderate (average accuracy between 58.7% and 75.7%). Has it been compared with other studies or methods? This comparison could be interesting. It could be interesting to use Machine Learning techniques that give very good results. I suggest that you consult, for example, the following paper, where results above 80% are achieved. (https://www.mdpi.com/1424-8220/18/6/1803 ).

RESPONSE: This is a very constructive suggestion. The method in this paper is based on the technology of random sampling to determine the size of sliding window, the biggest disadvantage is that the accuracy and calculation efficiency can’t meet at the same time: to improve accuracy, efficiency must be sacrificed; to improve efficiency, accuracy must be sacrificed. We also tried to find better ways to solve our problems, including new approaches proposed by reviewers such as machine learning and artificial intelligence. You provide the case of the paper, which is a quite good work. The comparison between various classification methods is remarkably complete and worth learning. Our work is a parameter optimization of outlier detection algorithm. We hope to have more new methods to compare and highlight the advantages and disadvantages of different methods as well. Since the work in this area is a continuation of the study in 2017, similar methods that can be referred to are rare. The proposal of new and advanced methods requires the design of new technical processes and the implementation of new work. It may require the writing of new algorithm programs and the implementation of a large number of experiments, which requires a certain amount of time to complete. For the exploration of new methods, we will continue to carry out in-depth research in the future. We also make a further prospect in the chapter of discussion and conclusion.

The details are as follows:

“Afore-mentioned experimental results explain, the obvious disadvantage of the new method is that the calculation accuracy and efficiency can’t be satisfied at the same time. With the rapid development and progress of computer computing power and artificial intelligence technology, it is an inevitable trend to apply the new method to the parameter optimization algorithm of sliding window size. For the exploration of new methods, we will continue to carry out in-depth research in the future work.”

COMMENT: 3) It would be interesting to describe the sample sizes that have been worked with. The total records processed to put in value the tests performed. In general I think that a good, structured and well-developed work has been done; however, I do not finish seeing the scientific novelty, I consider it not very relevant.

RESPONSE: It must be acknowledged that it is important to introduce the statistical characteristics of the modeling sample data. Therefore, we add the statistical attributes of average value, standard deviation and coefficient of variation in the data source information (Table 3). About the selection and analysis of the experimental test sample data, the article has made a detailed introduction. Monte Carlo method is an ancient and classic method, which can solve many complex practical problems and is also convenient for computer programming. Our work is a way to solve new problems by using classical methods.

COMMENT: Some minor aspects: I suggest leaving a blank line of space before and after the tables and some of the figures, they are very close to the text. For example:

Line 115: leave a blank line between 114 and 115. Also leave another free blank before 116.

Line 228: leave a blank line before it.

Line 257: leave a blank line before the figure. Same as Table 5 and Figure 5.

RESPONSE: We have modified according to your suggestion.

Thank you very much for your comments.

Sincerely yours,

Ni Jiang

Institution:  National Key Laboratory of Water Environmental Simulation and Pollution Control, South China Institute of Environmental Sciences, Ministry of Ecology and Environment

E-mail: jiangni@scies.org 
